# The Effect of Residual Architecture on the Per-Layer Gradient of Deep Networks

## Abstract

A critical part of the training process of neural networks takes place in the very first gradient steps post-initialization. In this work, we study the connection between the network's architecture and initialization parameters, to the statistical properties of the gradient in random fully connected ReLU networks, through the study of the the per layer Jacobian in finite sized networks. We compare three types of architectures: vanilla networks, ResNets and DenseNets. In our analysis, we show that while the variance of Jacobian squared norm is exponential in depth for ResNets, and polynomial for DenseNets, there exists an initialization strategy for both, such that the norm is preserved through arbitrary depths, preventing exploding or decaying gradients in deep networks. We also show that the statistics of the per layer Jacobian norm is a function of the architecture and the layer's size, but surprisingly, not the layer's depth.

## 1 Introduction

Understanding the effect of different architectures on the ability to train deep networks has long been a major topic of research. A recent line of work has focused on the statistical properties of both the activations and the gradients of different architectures at the point of initialization, in order to better understand the effect of different architectural choices on the training dynamics.

A well-known hurdle in the ability to train deep networks, is the effect of depth on the magnitude of the gradients. Specifically, the exploding and vanishing gradient phenomena have been observed in various deep architectures. they are considered a main culprit in the failure of training deep networks successfully. A flurry of architectural tricks and initialization techniques have been proposed, in order to mitigate the problem of exploding and vanishing gradients, ranging from residual architectures, different normalization techniques and orthogonal initializations.

In this work, we perform a rigorous analysis of the norm, of both the activations and the gradients of each layer, in fully connected architectures at the point of initialization. Our setting is as follows: we assume a fixed input $y^0$ to an $L$ layer neural network with weights given by $W$, and intermediate output for any layer $l$ denoted by $y^l$. Denoting by $w_{ij}^k$ the weights $ij$ of layer $k$, we are interested in the quantities:

$$\|y^l\|^2, \ \ \|J^k\|^2 = \sum_{ij} \|\frac{\partial y^L}{\partial w_{ij}^k}\|^2 \tag{1}$$

representing the norm of the activations of layer $l$ and Jacobian with respect to layer $k$. We consider these quantities random variables at initialization, and, therefore, devote our analysis to deriving expressions for the mean and variance of both $\|y^l\|^2$ and $\|J^k\|^2$, over the random initializations of the weights. Note that the full gradient requires an additional loss term which is independent of the architecture, and so our attention is focused on $J^k$. While most initialization techniques are designed to maintain the squared length of intermediate activations through the layers, the multiplication of i.i.d random weights along input-to-output paths in increasingly deep networks might lead to unbounded higher order statistics. In the extreme case, while the mean of activation and gradients remain constant in deep layers, it would be improbable to achieve mean values for any specific realization of the weights, rendering training difficult. Specifically, given some loss function $\mathcal{L}$, the

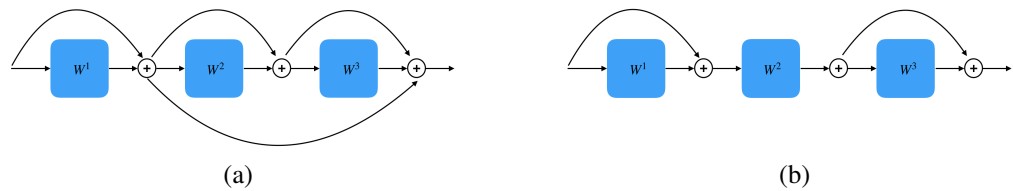

(a)             (b)

Figure 1: An illustration of Thm. 1. The activations of the network in (a) are completely different from those of the network in (b), in which all skip connections bypassing layer $k = 2$ are removed. However, the moments of the gradient norms at layer $k = 2$ are exactly the same in both (a) and (b).

norm of the gradient with respect to $W^k$ is given by:

$$\|\nabla_{W^k}\mathcal{L}\| = \|\frac{\partial \mathcal{L}}{\partial y^L}J^k\| \leq \|\frac{\partial \mathcal{L}}{\partial y^L}\|\|J^k\| \tag{2}$$

Large fluctuations to the Jacobian norm with respect to small perturbations of the weights will, therefore, cause wildly fluctuating gradients during training, hindering training dynamics.

Our analysis makes use of a duality theorem, which captures an intuitive concept. In residual architectures, there are many alternative paths of varying lengths from input to output. The theorem states that when considering the moments of the gradient at a certain layer, one can change the architecture, so all connections which that skip that layer are removed. This is true, despite the activations of the two networks being very different. See Fig. 1.

Our contributions are as follows: (i) We rigorously study the effect of different architectures, namely vanilla, residual and densely connected architectures, on the mean and variance of $\|y^l\|^2$ and $\|J^k\|^2$ in finite depth and width networks. (ii) From our forward-backward norm propagation duality theorem, we derive conditions that prevent exploding gradients in each of these architectures. (iii) We prove that for densely connected networks, the commonly used initialization of He et al. (2015) is enough to ensure the boundness of $var(\|J^k\|^2)$ and $var(\|y^l\|^2)$ for arbitrarily deep networks, shedding new light on the apparent advantage of these architectures in practical applications. (iv) We prove that replacing the standard conv-ReLU blocks with blocks of the concatenated ReLU (CR) activation presented by Shang et al. (2016), while maintaining the same parameter budget, significantly reduces the fluctuations of both the activations and gradients at initialization for any ReLU based architecture. We evaluate the applicability of this simple alteration on a large number of UCI datasets, and demonstrate the advantage for CR based networks over their standard counterparts.

## 2 RELATED WORK

The effect of architecture and depth on the gradients has been a topic of considerable interest recently, leading to several insightful research directions. A major line of work has been focused on the mean field approach, where the propagation of signals in random networks are analyzed in the regime of infinite width (Yang & Schoenholz (2017),Schoenholz et al. (2017),Xiao et al. (2018)). In that case, the pre-activations in any layer are approximated by a Gaussian distribution with moments depending on the specific layer. For ReLU networks (without residual connections), controlling the exponential explosion or decay of both the activations and gradients with depth is a simple matter of correctly scaling the initialization parameters. Indeed, it is shown in Schoenholz et al. (2017) that classical feed forward networks exhibit exponential behavior on average, when not initialized properly. For the gradients, a recursion relation is derived:

$$\mathbb{E}\left((\frac{\partial y^L}{\partial y_i^l})^2\right) = \mathbb{E}\left((\frac{\partial y^L}{\partial y_i^{l+1}})^2\right)\sigma_w^2 \int \frac{1}{\sqrt{2\pi}}e^{-\frac{z^2}{2}}\phi'(\sqrt{q^*}z)dz \tag{3}$$

where $\phi'$ the derivative of the activation function and $\sigma_w^2$ denotes the variance of the weights. By initializing with $\sigma_w^2 = \frac{2}{fan-in}$, and considering that for the ReLU activation function $\phi'(z) = \mathbb{1}_{z>0}$, we have that

$$\mathbb{E}((\frac{\partial y^L}{\partial y_i^l})^2) = \mathbb{E}((\frac{\partial y^L}{\partial y_i^{l+1}})^2). \tag{4}$$

Mean field analysis of residual networks is presented in Yang & Schoenholz (2017), where it is shown that ResNets exhibit sub-exponential, or even polynomial dynamics, depending on the activation function. While mean field theory captures the "average" dynamics of random neural networks, it does not take into account the scale of gradient fluctuations that are crucial to the stability of gradient descent, which is the focus of this work. A more refined analysis presented in Pennington et al. (2017), considers the full spectrum of the input output Jacobian of infinitely wide networks, given by:

$$J_{IO} = \frac{\partial y^L}{\partial y^0} \tag{5}$$

Exploding or vanishing gradients are then prevented by analyzing conditions that give rise to dynamical isometry, a state in which the squared eigenvalues of $J_{IO}$ are all concentrated around 1. In (Pennington et al., 2017) it is shown that fully connected ReLU networks are incapable of reaching dynamical isometry, as apposed to sigmoidal networks, while in (Pennington et al., 2018) it is shown that dynamical isometry is achieved in a universal manner for a variety of activation functions. Dynamical isometry is a much stronger condition than the Jacobian norm, since it requires access to the entire spectrum. However, in previous work, this is done using various forms of infinite width approximations. Instead, this work focuses on finite depth corrections to the statistics of various quantities concerning the squared Frobenius norm of the per-layer Jacobian (derivative of the output with respect to the weights), and can be viewed as a complementary approach to analyzing the dynamics at the start of training. Our results cannot be obtained using large width approximations, since we seek to characterise the effect of both depth and width on the Jacobian.

A recent line of work, more related to the analysis presented in this paper, deals with the statistical properties of the norm of activations and gradients in random finite width networks. Hanin & Rolnick (2018) tackle two failure modes that are caused by exponential explosion or decay of the norm of intermediate layers. It is shown that for random fully connected vanilla ReLU networks, the variance of the squared norm of the activations exponentially increases, even when initializing with the $\frac{2}{fan-in}$ initialization. For ResNets, this failure mode can be overcome by correctly rescaling the residual branches. However, it is not clear how such a rescaling affects the back propagation of gradients. In our work, we prove an equivalence relation between the forward- and back-propagation of norms in most random ReLU networks. In addition, we present an analysis of DenseNets, revealing a surprising norm preservation quality that is independent of depth.

Hanin (2018) explores the conditions at initialization that give rise to the exploding or vanishing gradient problem, by analyzing the individual entries of the input-output Jacobian. In the case of a single output network, the layer $k$ to output Jacobian is given by the row vector:

$$J(k \rightarrow L) = (\frac{\partial y^L}{\partial y_i^l}, \; i = 1...n_k) = \frac{\partial y^L}{\partial y^l} \tag{6}$$

with $n_k$ denoting the width of layer $k$. In our work, we take a more direct approach to analyzing the full expression for the Jacobian of random networks given by:

$$J_{ij}^k = \frac{\partial y^L}{\partial y^k} \frac{\partial y^k}{\partial w_{ij}^k} = \langle J(k \rightarrow L), \frac{\partial y^k}{\partial w_{ij}^k} \rangle. \tag{7}$$

Moreover, the analysis presented by Hanin (2018) does not directly carry over to ResNets or DenseNets.

## 3    PRELIMINARIES AND NOTATIONS

We make use of the following notations: $\mathcal{N}$ denotes a neural network architecture with intermediate outputs $y^l$ of layer $l$ for a fixed input $y^0$. $y_i^l$ denotes the $i$'th component of the vector $y^l$, and $n_1...n_L$ denote the width of the corresponding layers, with $n$ the length of the input vector. $\| \cdot \|^2$ is the squared Euclidean norm, and we assume $\|y^0\|^2 = 1$. We denote the weight matrix associated with layer $l$ by $W^l \in \mathbb{R}^{n_{l-1} \times n_l}$, with lower case letters $w_{ij}^l$ denoting the individual components of $W^l$. Additional superscripts $W^{lk}$ are used, when several weight matrices are associated with layer $l$, for instance in residual, CR and DenseNet architectures. Throughout the paper, we assume that the weights are sampled i.i.d from some symmetric distribution with moments $\mathbb{E}((w_{ij}^l)^m) = c_m^l$,

such that $\forall_{ijl}$, $\forall_l$, $c_1^l = 0, c_2^l = \frac{2\alpha}{n_{l-1}}$. We further identify the following property of a distribution $\Delta = 2(c_4^l - 3(c_2^l)^2)/n_{l-1}$ (for Gaussian distributions, $\Delta = 0$). The derivative of the output unit $y^L$ with respect to $w_{ij}^l$ is given by $J_{ij}^l = \frac{\partial y^L}{\partial w_{ij}^l} = \frac{\partial y^L}{\partial y^l} \frac{\partial y^l}{\partial w_{ij}^l}$, with $\|J^l\|^2 = \sum_{ij}(J_{ij}^l)^2$. We will be concerned in our analysis with the quantities $\mathbb{E}(\|y^l\|^2)$, $var(\|y^l\|^2)$, $\mathbb{E}(\|J^l\|^2)$ and $var(\|J^l\|^2)$, where the expectations are taken across the random sampling of the weights, given the fixed input $y^0$.

**Multi Pathway Architectures**   We compare a vanilla fully-connected network to ResNets He et al. (2016) and DenseNets Huang et al. (2016) to a network that is built with CR Shang et al. (2016) layers. For the standard vanilla fully-connected network, the output for of layer $l$ is given by

$$y^l = \phi(W^{l\top}y^{l-1}), \tag{8}$$

where $\phi()$ denotes the ReLU non-linearity.

For residual architectures, we study models with $m$ layer residual branches of the form

$$y^l = y^{l-1} + \mathcal{R}_m(y^{l-1}) \tag{9}$$

where:

$$\mathcal{R}_m(y^{l-1}) = W^{lm\top}...\phi(W^{l2\top}\phi(W^{l1\top}y^{l-1})). \tag{10}$$

We denote the $k$'th intermediate output of the $l$'th residual branch by

$$\tilde{y}^{l-1,k} = W^{lk\top}...\phi(W^{l2\top}\phi(W^{l1\top}y^{l-1})), \tag{11}$$

so that, for example

$$\mathcal{R}_m(y^{l-1}) = W^{lm\top}\tilde{y}^{l-1,m-1} = W^{lm\top}\phi(W^{lm-1\top}\tilde{y}^{l-1,m-2}). \tag{12}$$

Furthermore, since the last layer of any residual branch $W^{lm}$ does not end with a non-linearity, we assume $c_2^{lm} = \frac{1}{n}$. DenseNets were recently introduced, demonstrating faster training, as well as improved performance on several popular datasets. The main architectual features introduced by DenseNets include the connection of each layer output to all subsequent layers, using concatenation operations instead of summation, such that the weights of layer $l$ multiply the concatenation of the outputs $y^0...y^{l-1}$, and is of dimensions $W^l \in \mathbb{R}^{ln \times n}$. For our analysis, we break $W^l$ into $l-1$ weight matrices $W^{l1}...W^{ll-1} \in \mathbb{R}^{n \times n}$, so that the output of layer $l$ is given by $y^l = \phi(\sum_{l'=1}^{l-1} W^{ll'\top}y^{l'})$.

## 4   FORWARD-BACKWARD NORM PROPAGATION DUALITY

In this section, we introduce a link between the propagation of the norm of the activations, and the norm of the Jacobian in different layers in random ReLU networks of finite width. This link will then allow us to study the statistical properties of the gradient in general architectures incorporating residual connections and concatenations with relative ease. Specifically, we would like to establish a connection between the first and second moments of the squared norm of the output $\|y^L\|^2$, and those of the Jacobian norm $\|J^k\|^2$. The family of architectures we consider is restricted, so that the ReLU activations are only applied after multiplication with weights. Using a path-based notation, any unit $t$ of the output $y_t^L$ can be decomposed to paths that go through weight matrix $W^k$, denoted by $y_t^{L,k}$ (neglecting an additional superscript for ResNets and DenseNets), and paths that skip $W^k$, denoted by $\hat{y}_t^{L,k}$. Assuming the input $\forall_j$, $y_j^0 = 1$, we have:

$$y_t^L = y_t^{L,k} + \hat{y}_t^{L,k} = \sum_{k \in \gamma_t} \prod_{l=1}^{|\gamma_t|} w_{\gamma_t,l}^l z_{\gamma_t} + \sum_{k \notin \gamma_t} \prod_{l=1}^{|\gamma_t|} w_{\gamma_t,l}^l z_{\gamma_t} \tag{13}$$

where the summation is over paths from input to unit $t$ of the output, indexed by $\gamma_t$, with $|\gamma_t|$ denoting the length of the path indexed by $\gamma_t$. In non-residual networks, we have $|\gamma_t| = L$ and $\sum_{\gamma_t}$ sums over $\prod_{l=0}^{L-1} n_l$ paths. The term $\prod_{l=1}^{|\gamma_t|} w_{\gamma_t,l}^l z_{\gamma_t}$ denotes the product of weights along path $\gamma_t$, multiplied by a binary variable $z_{\gamma_t} \in [0,1]$, indicating whether path $\gamma_t$ is active. Finally, $\sum_{k \notin \gamma}$ indicates summation over all possible paths that do not include a weight from layer $k$. In all relevant architectures, the squared norm of the Jacobian entry $\|J_{ij}^k\|^2$ is given by:

$$\|J_{ij}^k\|^2 = \|\frac{\partial y^{L,k}}{\partial w_{ij}^k}\|^2 = \sum_{t=1}^{n_L} \left( \sum_{ijk \in \gamma_t} \frac{1}{w_{ij}^k} \prod_{l=1}^{|\gamma_t|} w_{\gamma_t,l}^l z_{\gamma_t} \right)^2 \tag{14}$$

That is, the derivative of output unit $t$ with respect to weight $w_{ij}^k$ is given by the sum of all paths $\gamma$ that contain $w_{ij}^k$ divided by $w_{ij}^k$. In order to link $J^k$ with $y^L$, we make the following definition:

**Definition 1.** *Reduced network: the outputs of a reduced network $\mathcal{N}_{(k)}$ denoted by $y_{(k)}^1...y_{(k)}^L$ given input $y^0$ are obtained by removing all connections bypassing weight $W^k$ from the network $\mathcal{N}$.*

Note that for vanilla networks, it holds that $\mathcal{N}_{(k)} = \mathcal{N}$, and $\forall_{0<l\leq L}, \ y_{(k)}^l = y^l$. The following Lemma links between the moments of $y_{(k)}^L$ and those of $y^L$:

**Lemma 1.** *For any architecture described in Sec. 3, it holds that for $m \in \{2, 4\}$:*

$$\mathbb{E}\Big(\|y_{(k)}^L\|^m\Big) = \mathbb{E}\Big(\|y^{L,k}\|^m\Big) \tag{15}$$

Note that if network $\mathcal{N}$ contains connections bypassing layer $k$, then for a specific realization of the weights, the equality $y_{(k)}^L = y^{L,k}$ does not hold in general, due to different activation patterns in $\mathcal{N}$ and $\mathcal{N}_{(k)}$ induced by additional residual connections. In other words, paths that are open in $\mathcal{N}$ (indicated by the variables $z_{\gamma_t}$) are not necessarily so in $\mathcal{N}_{(k)}$, and vice versa. Lem. 1, however, states that the moments of both are equal in the family of considered ReLU networks.
The following theorem relates the moments of $\|J^k\|^2$ with those of $\|y_{(k)}^L\|$:

**Theorem 1.** *For any architecture described in Sec. 3,the following hold:*

$$\mathbb{E}\Big(\|J^k\|^2\Big) = \frac{1}{c_2^k}\mathbb{E}\Big(\|y_{(k)}^L\|^2\Big), \quad \frac{var(\|y_{(k)}^L\|^2)}{c_4^k} \leq var(\|J^k\|^2) \leq \frac{var(\|y_{(k)}^L\|^2)}{(c_2^k)^2} \tag{16}$$

where $c_2^k$ and $c_4^k$ are the second and fourth moments of the weights in layer $k$. Thm. 1 indicates that we can compute the moments of the Jacobian norm of layer $k$, by computing the moments of the output of the reduced network $y_{(k)}^L$. Thm. 1 also reveals a surprising property of the gradients in general Relu networks. That is, when the weights are sampled from the same distribution in each layer, the gradient's magnitude for each layer are equal in expectation, and depend only on the output statistics.

## 4.1 Vanilla networks

In order to compare our results for ResNets and DenseNets, we first derive the moments of $y_{(k)}^l$ in our setting for vanilla architectures. Note that for vanilla networks, the reduced network matches the original, and so $\forall_{0<l\leq L}, \ y_{(k)}^l = y^l$.

**Lemma 2.** *For a vanilla ReLU neural network, as described in Sec, 3, it holds for $\alpha = 1$:*

$$C_1 \exp\left[\sum_{l=1}^{L-1} \frac{5 - |\Delta|}{2n_l}\right] - C_2 \leq var\Big(\|J^k\|^2\Big) \leq C_3 \exp\left[\sum_{l=1}^{L-1} \frac{5 + |\Delta|}{n_l}\right] - C_4 \tag{17}$$

*where $C_1, C_2, C_3, C_4$ are constants that do not depend on $L$.*

By setting $\alpha = 1$, the variance of the Jacobian squared norm $\|J^k\|^2$ will increase exponentially with $\sum_{l=1}^{L} \frac{5}{n_l}$, as noted by Hanin & Rolnick (2018). This can be mitigated by linearly increasing the width of each layer with depth, so that $n_l = nl$, resulting in constant asymptotic behavior $\exp\left[\sum_{l=1}^{L-1} \frac{1}{nl}\right] \sim L^{\frac{1}{n}}$. This, however, would require significant resources, since the weight matrices would grow quadratically with depth. For comparison, in DenseNets the variance is maintained with depth, while the number of weights increases linearly.

## 4.2 Residual networks

Residual networks have reintroduced the concept of bypass connections, allowing the training of deep and narrow models with relative ease. Hanin & Rolnick (2018) showed shown that the fluctuations of the norm of subsequent layers do not exponentially increase, provided that the weights are initialized,

such that the expected norm $\mathbb{E}(\|y^l\|^2)$ is bounded for any layer $l$. Such an initialization is not constant, and is dependant on the depth of the network $L$. It is also worth mentioning that the same depth dependent initialization for ResNets was found by Zhang et al. (2019) to be crucial for training convolutional ResNets without applying batchnorm, as introduced in the original formulation of ResNets. Recall that in our analysis, we consider residual branches with $m$ layers with weights $W^{kk'}$, $0 < k \le L$, $0 < k' \le m,$, and so we will analyze the variance of $\|J^{kk'}\|^2 = \sum_{ij} \left( \frac{\partial y^L}{\partial w_{ij}^{kk'}} \right)^2$ for $k' < m$. Unlike vanilla networks, the outputs of the reduced network $y_{(k)}^l$ for $l > k$ are not equal to those of the full network $y^l$, and are given by:

$$y_{(k)}^l = \left\{ \begin{array}{ll} \mathcal{R}_m(y^{l-1}) + y^{l-1} & l \ne k \\ \mathcal{R}_m(y^{l-1}) & l = k \end{array} \right. \tag{18}$$

The following theorem states our results for ResNets.

**Theorem 2.** *For a ResNet as in Sec, 3 with $m \ge 2$, it holds that for $0 < k \le L, 0 < k' < m$:*

$$C_1 \rho_-^m \exp\left[ \frac{1}{2} L \rho_-^m \right] - C_2 \mu^2 \le var\left( \|J^{kk'}\|^2 \right) \le C_3 \rho_+^m \exp\left[ 2 L \rho_+^{\frac{m}{2}} \right] - C_4 \mu^2 \tag{19}$$

*where $\rho_\pm = \alpha^2 (1 + \frac{5 \pm |\Delta|}{n})$, $C_1, C_2, C_3, C_4$ are constants that do not depend on $L, m$, and $\mu = \alpha^m \left( 1 + \alpha^m \right)^{L-1}$.*

The gradients of weights in residual branches are, therefore, sensitive to the depth of each residual branch (large $m$). However, unlike their vanilla counterparts, by scaling the residual branches, such that $L \rho_+^m$ is kept constant, ensures that both $\mathbb{E}(\|J\|^2)$ and $var(\|J\|^2)$ remain bounded as the overall depth of the network grows (increasing $L$). Note that a similar initialization was proposed in Zhang et al. (2019) for training ResNets without batchnorm, albeit for a more restricted class of networks, and without an explicit dependency on width. Our result indicates that for narrow and deep networks with residual connections, training without batchnorm requires scaling in a both depth and width dependent manner.

### 4.3 DENSELY CONNECTED NETWORKS

DenseNets have shown improvement over ResNets in terms of accuracy and speed of training on several notable image classification tasks, since their inception. As far as we can ascertain, there have been no theoretical studies on the causes of DenseNets success, or the implications on the training process of its architectural novelties, namely dense connections and feature reuse by concatenation of previous layers. In the following, it is shown that the DenseNet architecture, coupled with the standard, depth independent $\frac{2}{n_{l-1}}$ initialization, gives rise to well-behaving gradients in any depth. By the definition of DenseNets given in Sec. 3, the reduced architecture outputs $y_{(k)}^l$ are given by:

$$y^l(k) = \left\{ \begin{array}{ll} \phi(\sum_{l'=1}^{l-1} W^{ll'\top} y^{l'}) & l \le k \\ \phi(\sum_{l'=k}^{l-1} W^{ll'\top} y^{l'}) & l > k \end{array} \right. \tag{20}$$

The following theorem states our results for DenseNets:

**Theorem 3.** *For a DenseNet as described in Sec, 3, it holds that for $0 < k \le L$:*

$$L^{(\alpha-1)} \left( C_1 \exp\left[ \frac{-2|\Delta|L}{n} \right] - C_2 \right) \le var(\|J^k\|^2) \le L^{2(\alpha-1)} \left( C_3 \exp\left[ \frac{2|\Delta|L}{n} \right] - C_4 \right) \tag{21}$$

*where $C_1, C_2, C_3, C_4$ are constants that do not depend on $L$.*

It is worth noting the difference between DenseNets, and both vanilla and ResNet architectures. In vanilla architectures, the mean of the activation norm, as well as the gradient norm are exponential in $\alpha$. However, the variance of the same quantities is exponential in $\sum_{l=1}^L \frac{1}{n_l}$. That is, fixing the mean by correctly setting $\alpha = 1$ does not prevent the gradients and activations from exploding in deep networks. In ResNets, scaling of the residual branches in order to avoid exploding gradients

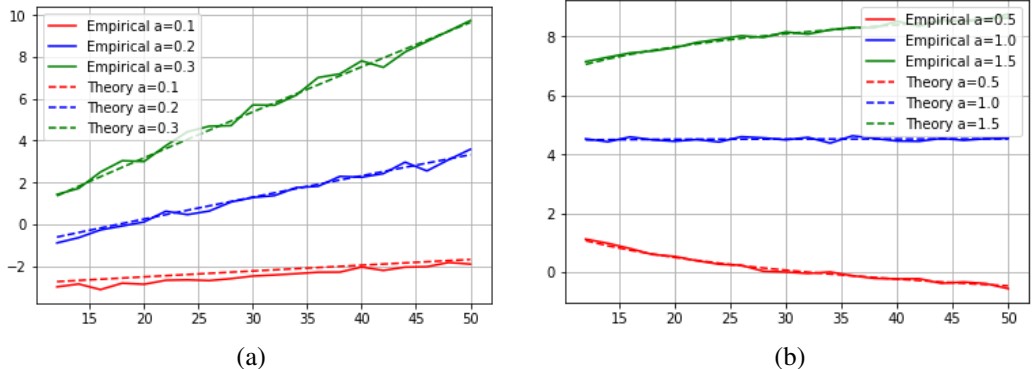

Figure 2: The variance of the squared norm in log scale of the per layer Jacobian as a function of $\alpha$ (held constant for all the layers) obtained from the simulated results of 200 independent runs, for (a) ResNet with $m = 2$ and $n = 40$, and (b) DenseNet with $n = 40$. The x-axis represents the depth of the network, and y-axis the variance as computed from 200 runs. The dashed lines represent our theoretical predictions for both architectures, and full lines represent the empirical results. All networks were initialized using Gaussian distributions.

requires scaling in a depth dependent fashion. However, in DenseNets, the variance of the squared Jacobian norm is in fact polynomial in depth for Gaussian distributions. Constant mean and variance is achieved (at least when $\Delta = 0$) by setting $\alpha = 1$, for any depth. Moreover, the constant variance result is achieved with a linear growth rate of the weight complexity, instead of quadratic with vanilla networks.

**Empirical support** We employed Monte Carlo simulations in order to verify our theoretical findings. As shown in Fig. 2, there is an excellent match between the simulations and our derivations.

## 5 THE CURIOUS CASE OF CONCATENATED RELU

Our results stated above demonstrate how different architectures and initialization schemes affect the training process at the start of training, by analyzing the expected norm and variance of the gradients for each layer, and hold for ReLU based architectures. Since a similar analysis can be performed on linear random networks, it is interesting to see the effect of the ReLU non-linearity on the outcome. As it turns out, the ReLU function has a detrimental effect on random networks, as shown in the following lemma, effectively exacerbating the fluctuations of both the activations and the gradients:

**Lemma 3.** *Given a vanilla linear neural network with $w_{ij}^l \sim \mathcal{N}(0, \frac{1}{n_{l-1}})$, then it holds that:*

$$\mathbb{E}\Big(\|y^L\|^2\Big) = \frac{n_L}{n}, \quad var\Big(\|y^L\|^2\Big) = \frac{n_L^2}{n^2}\Big(\prod_{l=1}^{L}(1 + \frac{2}{n_L}) - 1\Big) \tag{22}$$

Linear networks, of course, have limited capacity, considerably limiting their practical use. Fortunately, we can use the ReLU function to construct computation blocks that behave as linear networks at the initialization stage, while not compromising in terms of capacity. This can be achieved by the use of CR units. In concatenated ReLUs blocks, the input signal $x$ is split into two complementary components $x^+$ and $x^-$ such that the output of layer $l$ is parameterized by two weight matrices $W^{l1}, W^{l2}$, and is given by $y^l = W^{l1\top}\phi(y^{l-1}) - W^{l2\top}\phi(-y^{l-1})$. An algebraically equivalent way to write the layer's computation is $y^l = W^{l2\top}y^{l-1} + (W^{l1\top} + W^{l2\top})\phi(y^{l-1})$, which can be viewed as a variant of a residual connection. However, it is important to note that this equivalence holds for the forward-pass only, since deriving by $W^{l1}$ and $W^{l2}$ during the optimization process is not the same as performing an optimization that is based on a derivation by $W^{l1} - W^{l2}$ and $W^{l2}$. The equivalence at initialization between the CR based architectures and random linear networks is depicted in the following Lemma:

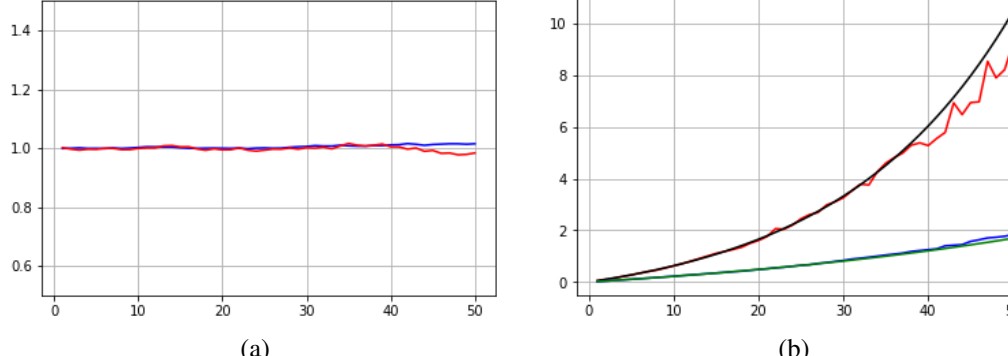

(a)                                                               (b)

Figure 3: Activations norm per layer, as obtained from simulated results of 5000 independent runs. (a) The mean of the squared norm of the activations in each layer (y-axis) as a function of depth (x-axis) for randomly initialized networks. The red plot is for a ReLU network, the blue for a CR based network. (b) the y-axis is the standard deviation of the squared norm of the activation. In addition to the blue and red plots, the black and green plots are the theoretical predictions according to Lem. 2,3.

**Lemma 4.** *The distribution of the output $y^l = W^{l1\top}\phi(y^{l-1}) - W^{l2\top}\phi(y^{l-1})$ conditioned on $y^{l-1}$, where $w^{l1}_{ij}, w^{l2}_{ij} \sim \mathcal{N}(0, \frac{\alpha}{n_{l-1}})$ equals the distribution of $y = W^{l\top}y^{l-1}$, where the elements of $W^l$ are distributed according to an i.i.d normal distribution $w^l_{ij} \sim \mathcal{N}(0, \frac{\alpha}{n_{l-1}})$.*

Using CR, therefore, provides initial conditions identical to random linear networks in terms of the output statistics. However, this equivalence also holds for back propagation. Indeed, from the construction of $W^l$ in the lemma, it is easy to see that $(\frac{\partial y^l}{\partial w^l_{ij}})^2 = (\frac{\partial y^l}{\partial w^{l1}_{ij}})^2 + (\frac{\partial y^l}{\partial w^{l2}_{ij}})^2$. A surprising property emerges. Lem. 4 shows that similar to vanilla ReLU based networks, CR based ReLU networks have exponential behavior in $\sum_{l=1}^{L} \frac{2}{n_l}$. Since CR employ twice as many parameters, then for the same budget of parameters (having $n_l = \frac{n_l}{\sqrt{2}}$ for CR) we have $\sum_{l=1}^{L} \frac{2\sqrt{2}}{n_l} < \sum_{l=1}^{L} \frac{5}{n_l}$, hence more efficient than standard ReLU blocks. We can similarly define residual and densely connected architecture comprised of concatenated ReLU blocks. In that case, the conditions of both architectures in terms of output distributions at initialization will be identical to the linear counterparts of both architectures, given by $y^l = W^{lk_1\top}\phi(\tilde{y}^{l,k-1}) + W^{lk_2\top}\phi(-\tilde{y}^{l,k-1})$ and $y^l = \sum_{l'=0}^{l-1} W^{ll'\top}y^{l'}$ for ResNets and DenseNets, respectively. Further evidence for the practical effectiveness of CR layers is provided in the appendix, in which we present experiments on the UCI datasets.

## 6   Conclusions

The interplay between architecture and initialization determines much of the early dynamics of training deep networks. In this work, we have demonstrated how careful initialization in general residual architectures effects the gradients at the start of training, by a rigorous study of the variance of the per layer gradient norm at the point of initialization. Through the use of the duality principle between forward and backward propagation, we have derived the expression for the variance in three architectures: standard networks, ResNets and DenseNets. We show that DenseNets benefit from a simple variance preserving initialization, which is independent of depth, while a depth dependent initialization exists for ResNet. We have also shown that CR blocks perform better then ReLU blocks, given a fixed budget of weights, when it comes to norm preservation in deep layers. As future work, we would like to extend our analysis to include normalization techniques, such as batchnorm.

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

# 7 PROOFS

We make use of the following propositions and definitions in order to prove lemma 1 and theorem 1. For ResNet, we assume $m = 2$ for notation simplicity. The results hold for $m > 2$ as well with no additional arguments.

A path $\gamma_t$ from input to output unit $t$ defines a product of weights along the path denoted by

$$Q_{\gamma_t} = \prod_{l=1}^{|\gamma_t|} w_{\gamma_t,l} z_{\gamma_t} = \prod_{l=1}^{|\gamma_t|} q_{\gamma_t,l} \tag{23}$$

where $|\gamma_t|$ is the length of path $\gamma_t$ (for vanilla and DenseNets $|\gamma_t| = L$), $z_{\gamma_t}$ is a binary variable $z_{\gamma_t} \in \{0,1\}$ indicating if the path is "active" (i.e, all relevant Relu activations along the path are open), and $w_{\gamma_t,l}$ are the weights along path $\gamma_t$, and $q_{\gamma_t,l} = w_{\gamma_t,l} z_{\gamma_t}$.

We generalize the above expression the following way:

$$Q_{\gamma_t} = \prod_{l=1}^{L} q_{\gamma_t,l} \tag{24}$$

where:

$$q_{\gamma_t,l} = \begin{cases} 1 & l \notin \gamma_t \\ \tilde{w}_{\gamma_t,l} z_{\gamma_t,l} & else \end{cases} \tag{25}$$

where $l \notin \gamma_t$ indicates if layer $l$ is skipped, $z_{\gamma_t,l}$ is the activation variable relevant for weight $\tilde{w}_{\gamma_t,l}$. That is, if $\tilde{w}_{\gamma_t,l} = w_{ij}^l$ then $z_{\gamma_t,l} = z_j^l$ where $z_j^l \in \{0,1\}$ is the activation variable unit $j$ in layer $l$. $\tilde{w}_{\gamma_t,l}$ denotes a generalized weight that can take the value of a weight $\tilde{w}_{\gamma_t,l} = w_{\gamma_t,l}$ for vanilla layers, or a product of weights $\tilde{w}_{\gamma_t,l} = w_{\gamma_t,l_1} w_{\gamma_t,l_2}$ for residual layers ($w_{\gamma_t,l_1} \in W^{l1}$ and $w_{\gamma_t,l_2} \in W^{l2}$).

**proposition 1.** *Given a random vector $w = [w_1...w_n]$ such that each component is identically and symmetrically distributed iid random variable with moments $\mathbb{E}(w_i^m) = c_m$, and random binary variable $z$ such that $z|w \in \{0,1\}$, $z|w = 1 - z| - w$, then it holds that for $n > l$:*

$$\mathbb{E}(\prod_{i=1}^{l} w_i^m z) = \begin{cases} 0 & m \text{ is odd, } l \text{ is even} \\ \frac{(c_m)^l}{2} & m \text{ is even} \\ \mathbb{E}(\prod_{i=1}^{l} w_i^m z) & else \end{cases} \tag{26}$$

*Proof.* We have:

$$c_m^l = \int_w \prod_{i=1}^{l} w_i^m p(w)dw = \int_{w|z=1} \prod_{i=1}^{l} w_i^m p(w)dw + \int_{w|z=0} \prod_{i=1}^{l} w_i^m p(w)dw \tag{27}$$

$$= \int_{w|z=1} \prod_{i=1}^{l} w_i^m p(w)dw + \int_{w|z=1} \prod_{i=1}^{l} (-w_i)^m p(w)dw \tag{28}$$

$$= \int_w \prod_{i=1}^{l} w_i^m z p(w)dw + \int_w \prod_{i=1}^{l} (-w_i)^m z p(w)dw \tag{29}$$

For even $m$ or even $l$: it follows that:

$$c_m^l = 2 \int_w \prod_{i=1}^{l} w_i^m z p(w)dw \tag{30}$$

and so:

$$\mathbb{E}(\prod_{i=1}^{l} w_i^m z) = \frac{c_m^l}{2} \tag{31}$$

For odd $l$, since $w$ is symmetrically distributed, we have that $c_m^l = 0$, and so $\mathbb{E}(\prod_{i=1}^{l} w_i^m z) = \frac{c_m^l}{2} = 0$ □

**proposition 2.** *For any set of paths $\gamma_t^1, \gamma_t^2$, it holds that:*

$$\mathbb{E}(Q_{\gamma_t^1} Q_{\gamma_t^2}) = \begin{cases} 0 & \gamma_t^1 \neq \gamma_t^2 \\ \mathbb{E}(Q_{\gamma_t^1}^2) = \prod_{l=1}^{L} \mathbb{E}\left((q_{\gamma_t^1,l})^2\right) & \gamma_t^1 = \gamma_t^2 \end{cases} \tag{32}$$

*Proof.* From proposition 2 we have:

$$\mathbb{E}(Q_{\gamma_t^1} Q_{\gamma_t^2}) = \mathbb{E}(\prod_{l=1}^{L} q_{\gamma_t^1,l} q_{\gamma_t^2,l}) \tag{33}$$

Since the computations done by all considered architectures form a markov chain, such that the output of layer $y^l$ depends only on weights $R^l = \{W^1...W^l\}$, using conditional expectation,:

$$\mathbb{E}(Q_{\gamma_t^1} Q_{\gamma_t^2}) = \mathbb{E}\left(E(Q_{\gamma_t^1} Q_{\gamma_t^2} \big| R^{L-1})\right) = \mathbb{E}\left(\prod_{l=1}^{L-1} q_{\gamma_t^1,l} q_{\gamma_t^2,l} E(q_{\gamma_t^1,L} q_{\gamma_t^2,L} \big| R^{L-1})\right) \quad (34)$$

Note that since both are paths leading to output unit $t$, the following holds:

$$E(q_{\gamma_t^L,l} q_{\gamma_t^2,L} \big| R^{L-1}) = \begin{cases} \mathbb{E}(\tilde{w}_{\gamma_t^1,L} \tilde{w}_{\gamma_t^2,L} z_t^L) & L \in \gamma_t^1, L \in \gamma_t^2 \\ \mathbb{E}(\tilde{w}_{\gamma_t^1,L} z_t^L) & L \in \gamma_t^1, L \notin \gamma_t^2 \\ \mathbb{E}(\tilde{w}_{\gamma_t^2,L} z_t^L) & L \notin \gamma_t^1, L \in \gamma_t^2 \\ 1 & L \notin \gamma_t^1, L \notin \gamma_t^2 \end{cases} \quad (35)$$

and so it follows from proposition 1:

$$E(q_{\gamma_t^L,l} q_{\gamma_t^2,L} \big| R^{L-1}) = \begin{cases} \mathbb{E}(\tilde{w}_{\gamma_t^1,L}^2 z_t^L) & L \in \gamma_t^1, L \in \gamma_t^2, \tilde{w}_{\gamma_t^1,L} = \tilde{w}_{\gamma_t^2,L} \\ 1 & L \notin \gamma_t^1, L \notin \gamma_t^2 \\ 0 & else \end{cases} = \mathbb{E}\left((q_{\gamma_t^1,L})^2\right) \quad (36)$$

Recursively going through $l = l - 1...1$ completes the proof. $\qquad \square$

**proposition 3.** *For any set of paths $\gamma_t^1, \gamma_t^2, \gamma_s^1, \gamma_s^2$, from input to output units $t$ and $s$, it holds that:*

$$\mathbb{E}(Q_{\gamma_t^1} Q_{\gamma_t^2} Q_{\gamma_s^1} Q_{\gamma_s^2}) = \prod_{l=1}^{L} \mathbb{E}\left(\xi_{\gamma_{st}^{12},l}\right) \quad (37)$$

*such that:*

$$\mathbb{E}\left(\xi_{\gamma_{st}^{12},l}\right) = \begin{cases} E\left(q_{\gamma_t^1,l}^2 q_{\gamma_s^1,l}^2\right) & q_{\gamma_t^1,l} = q_{\gamma_t^2,l}, q_{\gamma_s^1,l} = q_{\gamma_s^2,l} \\ E\left(q_{\gamma_t^1,l}^2 q_{\gamma_t^2,l}^2\right) & q_{\gamma_t^1,l} = q_{\gamma_s^1,l}, q_{\gamma_t^2,l} = q_{\gamma_s^2,l} \\ 0 & else \end{cases} \quad (38)$$

*Proof.* The proof is practically identical to proposition 3, and will be omitted. $\qquad \square$

*For notation simplicity,*

## 7.1 PROOF OF LEMMA 1

The proof of lemma 1 stems from the following observation: Since the computations done by all considered architectures form a markov chain, such that the output of layer $y^l$ depend only weights $W^1...W^l$, then for any particular realization of the weights, $y^{L,k}$ and $y^L k$ differ only in their activation patterns in layers $l \geq k$ due to the removal of bypassing connections in $y^L(k)$. Using Eq. 23:

$$\mathbb{E}\left(\|y^{L,k}\|^m\right) = \sum_{t=1}^{n_L} \mathbb{E}\left(\left(\sum_{\gamma_t | k \in \gamma_t} \prod_{l=1}^{|\gamma_t|} w_{\gamma_t,l} z_{\gamma_t}\right)^m\right) \quad (39)$$

while:

$$\mathbb{E}\left(\|y_{(k)}^L\|^m\right) = \sum_{t=1}^{n_L} \mathbb{E}\left(\left(\sum_{\gamma_t | k \in \gamma_t} \prod_{l=1}^{|\gamma_t|} w_{\gamma_t,l} z'_{\gamma_t}\right)^m\right) \quad (40)$$

We must therefor prove the expectation is invariant to a switch between $z_{\gamma_t}$ and $z'_{\gamma_t}$ for layers $l > k$. For $m = 2$, using proposition 2, the expectation factorizes:

$$\mathbb{E}\left(\|y^{L,k}\|^2\right) = \sum_{t=1}^{n_L} \sum_{\gamma_t^1 | k \in \gamma_t} \mathbb{E}(Q_{\gamma_t^1}^2) = \sum_{t=1}^{n_L} \sum_{\gamma_t^1 | k \in \gamma_t} \prod_{l=1}^{L} \mathbb{E}\left((q_{\gamma_t^1,l})^2\right) \quad (41)$$

Similarly for $m = 4$, from proposition 3 the expectation factorizes:

$$\mathbb{E}\left(\|y^{L,k}\|^4\right) = \sum_{t=1}^{n_L} \sum_{k\in\gamma_t^1,\gamma_t^2,\gamma_s^1,\gamma_s^2} \prod_{l=1}^{L} \mathbb{E}\left(\xi_{\gamma_{st}^{12},l}\right) \tag{42}$$

where $\xi_{\gamma_{st}^{12},l}$ is defined is Eq. 38. Note that switching the sign of $W^l$ for all architectures considered will switch the state of both $z_j^l, 0 < j \le n_l$ and $z_j'^l, 0 < j \le n_l$, and so from proposition 1, the factorized expectations for $m = 2$ and $m = 4$ meet the required invariance.

## 7.2 PROOF OF THEOREM 1

From the definition of the Jacobian for weights $W^k$, we have using lemma 1 and proposition 2:

$$\mathbb{E}(\|J_{ij}^k\|^2) = \mathbb{E}(\|\frac{\partial y_{(k)}^L}{\partial w_{ij}^k}\|^2) = \sum_{t=1}^{n_L} \sum_{ijk\in\gamma_t^1,\gamma_t^2} \mathbb{E}\left(\frac{1}{(w_{ij}^k)^2} \prod_{l=1}^{L} Q_{\gamma_t^1} Q_{\gamma_t^2}\right)$$

$$= \sum_{t=1}^{n_L} \sum_{ijk\in\gamma_t} \mathbb{E}\left(\frac{1}{(w_{ij}^k)^2} \prod_{l=1}^{L} Q_{\gamma_t}^2\right) \tag{43}$$

Since by definition $w_{ij}^k \in \gamma_t^1$, from proposition 2 the term $\mathbb{E}\left(\frac{1}{(w_{ij}^k)^2} \prod_{l=1}^{L} Q_{\gamma_t}^2\right)$ factorizes:

$$\mathbb{E}\left(\frac{1}{(w_{ij}^k)^2} \prod_{l=1}^{L} Q_{\gamma_t}^2\right) = \mathbb{E}(\frac{q_{\gamma_t,k}^2}{(w_{ij}^k)^2}) \prod_{l\neq k,l=1}^{L} \mathbb{E}(q_{\gamma_t,l}^2) = \mathbb{E}(\frac{(w_{ij}^k)^2 z_i^k}{(w_{ij}^k)^2}) \prod_{l\neq k,l=1}^{L} \mathbb{E}(q_{\gamma_t,l}^2) \tag{44}$$

$$= \mathbb{E}(z_i^k) \prod_{l\neq k,l=1}^{L} \mathbb{E}(q_{\gamma_t,l}^2) = \frac{1}{2} \prod_{l\neq k,l=1}^{L} \mathbb{E}(q_{\gamma_t,l}^2) = \mathbb{E}\frac{1}{c_2^k}\left(\prod_{l=1}^{L} Q_{\gamma_t}^2\right) \tag{45}$$

where the last transition stems from proposition 1. Plugging back into Eq. 43 and summing over $ij$ completes the proof for $m = 2$.
For $m = 4$, we have:

$$\mathbb{E}(\|J^k\|^4) = \mathbb{E}(\sum_{ijuv} \|\frac{\partial y_{(k)}^L}{\partial w_{ij}^k}\|^2 \|\frac{\partial y_{(k)}^L}{\partial w_{uv}^k}\|^2)$$

$$= \sum_{t=1}^{n_L} \sum_{s=1}^{n_L} \sum_{ijk\in\gamma_t^1,\gamma_t^2} \sum_{uvk\in\gamma_s^1,\gamma_s^2} \mathbb{E}\left(\frac{1}{(w_{ij}^k)^2(w_{uv}^k)^2} \prod_{l=1}^{L} Q_{\gamma_t^1} Q_{\gamma_t^2} Q_{\gamma_s^1} Q_{\gamma_s^2}\right) \tag{46}$$

From proposition 3, for $uvk\in\gamma_s^1,\gamma_s^2, ijk\in\gamma_t^1,\gamma_t^2$ the expectation factorizes:

$$\mathbb{E}\left(\frac{1}{(w_{ij}^k)^2(w_{uv}^k)^2} \prod_{l=1}^{L} Q_{\gamma_t^1} Q_{\gamma_t^2} Q_{\gamma_s^1} Q_{\gamma_s^2}\right) = \mathbb{E}\left(\frac{\xi_{\gamma_{st}^{12},k}}{(w_{ij}^k)^2(w_{uv}^k)^2}\right) \prod_{l\neq k,l=1}^{L} \mathbb{E}\left(\xi_{\gamma_{st}^{12},l}\right) \tag{47}$$

$$\tag{48}$$

where from proposition 1 and proposition 3, for $uvk\in\gamma_s^1,\gamma_s^2, ijk\in\gamma_t^1,\gamma_t^2$:

$$\mathbb{E}\left(\frac{\xi_{\gamma_{st}^{12},k}}{(w_{ij}^k)^2(w_{uv}^k)^2}\right) = \begin{cases} \frac{\mathbb{E}(\xi_{\gamma_{st}^{12},k})}{(c_2^k)^2} & \{u,v\} \neq \{i,j\} \\ \frac{\mathbb{E}(\xi_{\gamma_{st}^{12},k})}{c_4^k} & \{u,v\} = \{i,j\} \end{cases} \tag{49}$$

Note that since $c_4^k \ge (c_2^k)^2$, it holds that:

$$\frac{\mathbb{E}(\xi_{\gamma_{st}^{12},k})}{c_4^k} \le \mathbb{E}\left(\frac{\xi_{\gamma_{st}^{12},k}}{(w_{ij}^k)^2 w_{uv}^k)^2}\right) \le \frac{\mathbb{E}(\xi_{\gamma_{st}^{12},k})}{(c_2^k)^2} \tag{50}$$

Plugging into Eq. 46, it follows:

$$\frac{var(\|y_{(k)}^L\|^2)}{c_4^k} \leq var(\|J^k\|^2) = \mathbb{E}(\|J^k\|^4) - \mathbb{E}(\|J^k\|^2)^2 \leq \frac{var(\|y_{(k)}^L\|^2)}{(c_2^k)^2} \tag{51}$$

Furthermore, it holds that:

$$\mathbb{E}(\|J^k\|^4) = \sum_{st} \sum_{ijk\in\gamma_t^1,\gamma_t^2} \sum_{uvk\in\gamma_s^1,\gamma_s^2} \mathbb{E}\left(\frac{\xi_{\gamma_{st}^{12},k}}{(w_{ij}^k)^2(w_{uv}^k)^2}\right) \prod_{l\neq k,l=1}^{L} \mathbb{E}\left(\xi_{\gamma_{st}^{12},l}\right) \tag{52}$$

$$= \sum_{st} \sum_{ijk\in\gamma_t^1,\gamma_t^2} \sum_{uvk\in\gamma_s^1,\gamma_s^2} \mathbb{E}\left(\frac{\xi_{\gamma_{st}^{12},k}}{(w_{ij}^k)^2(w_{uv}^k)^2}\right)\left(\mathbb{1}_{\{u,v\}=\{i,j\}} + \mathbb{1}_{\{u,v\}\neq\{i,j\}}\right) \prod_{l\neq k,l=1}^{L} \mathbb{E}\left(\xi_{\gamma_{st}^{12},l}\right) \tag{53}$$

Therefor for large width networks:

$$\mathbb{E}(\|J^k\|^4) \sim \sum_{st} \sum_{ijk\in\gamma_t^1,\gamma_t^2} \sum_{uvk\in\gamma_s^1,\gamma_s^2} \mathbb{E}\left(\frac{\xi_{\gamma_{st}^{12},k}}{(w_{ij}^k)^2(w_{uv}^k)^2}\right) \mathbb{1}_{\{u,v\}\neq\{i,j\}} \prod_{l\neq k,l=1}^{L} \mathbb{E}\left(\xi_{\gamma_{st}^{12},l}\right) \tag{54}$$

$$= \frac{1}{(c_2^k)^2} \sum_{st} \sum_{ijk\in\gamma_t^1,\gamma_t^2} \sum_{uvk\in\gamma_s^1,\gamma_s^2} \mathbb{E}\left(\xi_{\gamma_{st}^{12},k}\right) \mathbb{1}_{\{u,v\}\neq\{i,j\}} \prod_{l\neq k,l=1}^{L} \mathbb{E}\left(\xi_{\gamma_{st}^{12},l}\right) \tag{55}$$

$$\sim \frac{1}{(c_2^k)^2} \mathbb{E}(\|y_{(k)}^L\|^4) \tag{56}$$

and so for large width networks:

$$var(\|J^k\|^2) \sim \frac{var(\|y_{(k)}^L\|^2)}{(c_2^k)^2} \tag{57}$$

### 7.3   PROOF OF LEMMA 2

Denoting by $Z^l$ the diagonal matrix holding in its diagonal the activation variables $z_j^l$ for unit $j$ in layer $l$, and so we have:

$$y^l = Z^l W^{l\top} y^{l-1} \tag{58}$$

Conditioning on $R^{l-1} = \{W^1...W^{L-1}\}$ and taking expectation:

$$\mathbb{E}\left(\|y^L\|^2|R^{L-1}\right) = y^{L-1\top} \mathbb{E}\left(W^L Z^L W^{L\top}\right) y^{L-1} \tag{59}$$

$$= \sum_{j=1}^{n_L} \sum_{i_1,i_2=1}^{n_{L-1}} y_{i_1}^{l-1} y_{i_2}^{l-1} \mathbb{E}\left(w_{i_1,j}^L w_{i_2,j}^L z_j^L|R^{L-1}\right) \tag{60}$$

From proposition 1, it follows that:

$$\mathbb{E}\left(\|y^L\|^2|R^{L-1}\right) = \frac{n_L}{n_{L-1}} \mathbb{E}\left(\|y^{L-1}\|^2\right) \tag{61}$$

Recursing through $l = L - 1...1$:

$$\mathbb{E}\left(\|y^L\|^2\right) = \frac{n_L}{n} \tag{62}$$

Similarly:

$$\mathbb{E}\left(\|y^L\|^4|R^{L-1}\right) = \mathbb{E}\left((y^{L-1\top} W^L Z^L W^{L\top} y^{L-1})^2\right) \tag{63}$$

$$= \sum_{j_1,j_2,i_1,i_2,i_3,i_4} y_{i_1}^{L-1} y_{i_2}^{L-1} y_{i_3}^{L-1} y_{i_4}^{L-1} \mathbb{E}\left(w_{i_1,j_1}^L w_{i_2,j_1}^L w_{i_3,j_2}^L w_{i_4,j_2}^L z_{j_1}^L z_{j_2}^L|R^{L-1}\right) \tag{64}$$

From proposition 3:

$$\mathbb{E}\left(w_{i_1,j_1}^L w_{i_2,j_1}^L w_{i_3,j_2}^L w_{i_4,j_2}^L z_{j_1}^L z_{j_2}^L | R^{L-1}\right) \quad (65)$$

$$= \mathbb{E}\left(w_{i_1,j_1}^L w_{i_2,j_1}^L w_{i_3,j_2}^L w_{i_4,j_2}^L z_{j_1}^L z_{j_2}^L | R^{L-1}\right)\left(\mathbb{1}_{j_1=j_2,i_1=i_2=i_3=i_4} + 3\mathbb{1}_{j_1=j_2,i_1=i_2,i_3=i_4,i_1\neq i_3}\cdots \quad (66)\right.$$

$$\left. +\mathbb{1}_{j_1\neq j_2,i_1=i_2,i_3=i_4}\right) \quad (67)$$

and so:

$$\mathbb{E}\left(\|y^L\|^4\right) = \frac{n_L c_4^L}{2}\sum_{i=1}\mathbb{E}\left((y_i^{L-1})^4\right) + \frac{6n_L}{n_{L-1}^2}\sum_{i_1\neq i_2}\mathbb{E}\left((y_{i_1}^{L-1})^2(y_{i_2}^{L-1})^2\right) \quad (68)$$

$$+\frac{n_L(n_L-1)}{n_{L-1}^2}\sum_{i_1,i_2}\mathbb{E}\left((y_{i_1}^{L-1})^2(y_{i_2}^{L-1})^2\right) \quad (69)$$

$$= \frac{n_L(c_4^L - 3(c_c^L)^2)}{2}\sum_i\mathbb{E}\left((y_i^{L-1})^4\right) + \frac{n_L(n_L+5)}{n_{L-1}^2}\mathbb{E}\left(\|y^{L-1}\|^4\right) \quad (70)$$

$$= \frac{n_L\Delta}{n_{L-1}^2}\sum_i\mathbb{E}\left((y_i^{L-1})^4\right) + \frac{n_L(n_L+5)}{n_{L-1}^2}\mathbb{E}\left(\|y^{L-1}\|^4\right) \quad (71)$$

finally:

$$\frac{n_L(n_L+5+\Delta)}{n_{L-1}^2}\mathbb{E}\left(\|y^{L-1}\|^4\right) \leq \mathbb{E}\left(\|y^L\|^4\right) \quad (72)$$

$$\leq \frac{n_L(n_L+5+|\Delta|)}{n_{L-1}^2}\mathbb{E}\left(\|y^{L-1}\|^4\right) \quad (73)$$

Recursing through $L-1...1$, and denoting $C_1 = \frac{n_L(n_L+5-|\Delta|)}{n^2}$, and $C_2 = \frac{n_L(n_L+5+|\Delta|)}{n^2}$ and combining with Eq. 62:

$$C_1\prod_{l=1}^{L-1}(1 + \frac{5-|\Delta|}{n_l}) - \frac{n_L^2}{n^2} \leq var\left(\|y^L\|^2\right) \leq C_2\prod_{l=1}^{L-1}(1 + \frac{5+|\Delta|}{n_l}) - \frac{n_L^2}{n^2} \quad (74)$$

Using $\forall_{1\geq x\geq -1}, \ e^{0.5x} \leq (1+x) \leq e^x$:

$$C_1\exp\left[\sum_{l=1}^{L}\frac{5-|\Delta|}{2n_l}\right] - C_2 \leq var\left(\|y^L\|^2\right) \leq C_3\exp\left[\sum_{i=1}^{L}\frac{5+|\Delta|}{n_l}\right] - C_4 \quad (75)$$

For Gaussian distribution we have that $\Delta = 0$, and so:

$$var\left(\|y^L\|^2\right) = \frac{n_L(n_L+5)}{n^2}\prod_{l=1}^{L-1}(1 + \frac{5}{n_l}) - \frac{n_L^2}{n^2} \quad (76)$$

### 7.4 PROOF OF THEOREM 2

In the following proof, for the sake of notation simplicity, we omit the notation $k$ in $y_{(k)}^l$, and assume that $y^l$ stands for the reduced network $y_{(k)}^l$. We have for layer $L$:

$$\mathbb{E}\left(\|y^L\|^2\right) = \mathbb{E}\left(\|y^{L-1}\|^2\right) + \mathbb{E}\left(\tilde{y}^{L-1,m-1\top}W^{Lm}W^{Lm\top}\tilde{y}^{L-1,m-1}\right)$$

$$= \mathbb{E}\left(\|y^{L-1}\|^2\right) + \alpha\mathbb{E}\left(\|\tilde{y}^{L-1,m-1}\|^2\right) = \mathbb{E}\left(\|y^{L-1}\|^2\right)\left(1 + \alpha^m\right)$$

$$= \mathbb{E}\left(\|y^k\|^2\right)\prod_{l=k+1}^{L}\left(1 + \alpha^m\right) = \mathbb{E}\left(\|y^{k-1}\|^2\right)\prod_{l=k+1}^{L}\left(1 + \alpha^m\right)\alpha^m$$

$$= \prod_{l\neq k}\left(1 + \alpha^m\right)\alpha^m \quad (77)$$

$$\mathbb{E}\left(\|y^L\|^4\right) = \mathbb{E}\left(\|y^{L-1}\|^4\right) + \mathbb{E}\left(\|\tilde{y}^{L-1,m}\|^4\right) + 4\mathbb{E}\left((\tilde{y}^{L-1,m\top}y^{L-1})^2\right)$$

$$+ 2\mathbb{E}\left(\|\tilde{y}^{L-1,m}\|^2\|y^{L-1}\|^2\right) \quad (78)$$

We now handle each term separately:

$$\mathbb{E}\left(\|\tilde{y}^{L-1,m}\|^4\right) = \mathbb{E}\left(\mathbb{E}\left(\|\tilde{y}^{L-1,m}\|^4|R^{L-1}\right)\right) \quad (79)$$

Similarly to the vanilla case with a linear weight layer on top, we have:

$$\alpha^{2m}(1 + \frac{5 - |\Delta|}{n})^{m-1}\|y^{L-1}\|^4 \leq \mathbb{E}\left(\|\tilde{y}^{L-1,m}\|^4|R^{L-1}\right) \quad (80)$$

$$\leq \alpha^{2m}(1 + \frac{5 + |\Delta|}{n})^{m}\|y^{L-1}\|^4 \quad (81)$$

$$\mathbb{E}\left((\tilde{y}^{L-1,m-1\top}y^{L-1})^2\right) = \sum_{j_1 j_2 i_1 i_2} \mathbb{E}\left(\tilde{y}_{i_1}^{L-1,m-1}\tilde{y}_{i_2}^{L-1,m-1}y_{j_1}^{L-1}y_{j_2}^{L-1}w_{i_1,j_1}^{Lm}w_{i_2,j_2}^{Lm}\right) \quad (82)$$

$$= \frac{\alpha}{n}\mathbb{E}\left(\|\tilde{y}^{L-1,m-1}\|^2\|y^{L-1}\|^2\right) = \frac{\alpha^m}{n}\mathbb{E}\left(\|y^{L-1}\|^4\right) \quad (83)$$

and:

$$\mathbb{E}\left(\|\tilde{y}^{L-1,m}\|^2\|y^{L-1}\|^2\right) = \alpha^m\mathbb{E}\left(\|y^{L-1}\|^4\right) \quad (84)$$

Plugging it all into Eq. 78, and denoting:

$$\beta_l^+ = 1 + 2\alpha^m(1 + \frac{2}{n}) + \alpha^{2m}(1 + \frac{5 + |\Delta|}{n})^m \quad (85)$$

$$\beta_l^- = 1 + 2\alpha^m(1 + \frac{2}{n}) + \alpha^{2m}(1 + \frac{5 - |\Delta|}{n})^{m-1} \quad (86)$$

we have after recursing through $l = L - 1...1$:

$$\mathbb{E}\left(\|y^k\|^4\right)\prod_{l=k+1}^{L}\beta_l^- \leq \mathbb{E}\left(\|y^L\|^4\right) \leq \mathbb{E}\left(\|y^k\|^4\right)\prod_{l=k+1}^{L}\beta_l^+ \quad (87)$$

Since for the reduced architecture $y_{(k)}^L$, the transformation between $y^{k-1}$ and $y^k$ is given by an $m$ layer fully connected network without any residual connections, we can use the results from the vanilla case:

$$\prod_{l\neq k}^{L}\beta_l^-\alpha^{2m}(1 + \frac{5 - |\Delta|}{n})^{m-1} \leq \mathbb{E}\left(\|y^L\|^4\right) \leq \prod_{l\neq k}^{L}\beta_l^+\alpha^{2m}(1 + \frac{5 + |\Delta|}{n})^m \quad (88)$$

Denoting $\rho_\pm = \alpha^2(1 + \frac{5\pm|\Delta|}{n})$ it follows:

$$C_1\prod_{l\neq k}^{L}\beta_l^-\rho_-^m \leq \mathbb{E}\left(\|y^L\|^4\right) \leq \prod_{l\neq k}^{L}\beta^+\rho_+^m \quad (89)$$

From the definition of $\beta_l^\pm$, it holds for $\rho_+ < 1$:

$$\beta_l^+ \leq \left(1 + \rho_+^{\frac{m}{2}}\right)^2 \leq \exp\left[2\rho_+^{\frac{m}{2}}\right] \quad (90)$$

and:

$$\beta_l^- \geq \left(1 + \rho_-^m\right) \geq \exp\left[\frac{1}{2}\rho_-^m\right] \quad (91)$$

and so:

$$C_1\rho_-^m\exp\left[\frac{1}{2}L\rho_-^m\right] - C_2\mu^2 \leq var\left(\|y^L\|^2\right) \leq C_3\rho_+^m\exp\left[2L\rho_+^{\frac{m}{2}}\right] - C_3\mu^2 \quad (92)$$

For Gaussian distributions, using a similar derivation and assigning $\Delta = 0$ we have:

$$\mathbb{E}\left(\|y^L\|^4\right) = \rho \prod_{l \neq k}^{L} \beta_l \tag{93}$$

where:

$$\beta_l = 1 + 2\alpha^m(1 + \frac{2}{n}) + \alpha^{2m}(1 + \frac{5}{n})^{m-1}(1 + \frac{2}{n}) \tag{94}$$

and:

$$\rho = a^{2m}(1 + \frac{5}{n})^{m-1}(1 + \frac{2}{n}) \tag{95}$$

### 7.5 PROOF OF THEOREM 3

In the following proof, for the sake of notation simplicity, we omit the notation $k$ in $y^l_{(k)}$, and assume that $y^l$ stands for the reduced network $y^l_{(k)}$. We have:

$$\mathbb{E}\left(\|y^L\|^2\right) = \mu_L = \mathbb{E}\left((\sum_{l=k}^{L-1} y^{l\top}W^{Ll})Z^L(\sum_{l=k}^{L-1} y^{l\top}W^{Ll})\right) = \frac{a_L}{L}\sum_{l=k}^{L-1}\mu_{l-1} \tag{96}$$

$$\mathbb{E}\left(\|y^L\|^4\right) = \mathbb{E}\left(\left(\sum_{l=k}^{L-1}(y^{l\top}W^{Ll})Z^L\sum_{l=k}^{L-1}(y^{l\top}W^{Ll})\right)^2\right)$$

$$= \mathbb{E}\left(\left(\sum_{l_1=k}^{L-1}(y^{l_1\top}W^{Ll_1})Z^L\sum_{l_2=k}^{L-1}(y^{l_2\top}W^{Ll_2})\sum_{l_3=k}^{L-1}(y^{l_3\top}W^{Ll_3})Z^L\sum_{l_4=k}^{L-1}(y^{l_4\top}W^{Ll_4})\right)\right) \tag{97}$$

Note that the above is a simple Relu block with concatenated $y^1...y^L - 1$ as input. Using the results from the vanilla architecture, and denoting $C_{l,l'} = \mathbb{E}\left(\|y^l\|^2\|y^{l'}\|^2\right)$, it then follows:

$$\frac{(n + 5 - |\Delta|)\alpha^2}{nL^2}\sum_{l_1,l_2=k}^{L-1}C_{l_1,l_2} \leq C_{L,L} \leq \frac{(n + 5 + |\Delta|)\alpha^2}{nL^2}\sum_{l_1,l_2=k}^{L-1}C_{l_1,l_2} \tag{98}$$

From Eq. 98, it holds that:

$$\frac{n(L-1)^2}{n+5+|\Delta|}C_{L-1,L-1} \leq \sum_{l_1,l_2=k}^{L-2}C_{l_1,l_2} \leq \frac{n(L-1)^2}{n+5-|\Delta|}C_{L-1,L-1} \tag{99}$$

It then follows:

$$\mathbb{E}(\|y^L\|^4) = C_{L,L} \leq \frac{\alpha^2}{L^2}(1 + \frac{5+|\Delta|}{n})\sum_{l_1,l_2=k}^{L-1}C_{l_1l_2}$$

$$= \frac{\alpha^2}{L^2}(1 + \frac{5+|\Delta|}{n})\left(C_{L-1,L-1} + \sum_{l_1,l_2=k}^{L-2}C_{l_1l_2} + 2\sum_{l=k}^{L-2}C_{L-1,l}\right)$$

$$= \frac{\alpha^2}{L^2}(1 + \frac{5+|\Delta|}{n})\left(C_{L-1,L-1} + \frac{(L-1)^2n}{a_{L-1}^2(n+5-|\Delta|)}C_{L-1,L-1} + 2\sum_{l=k}^{L-2}C_{L-1,l}\right) \tag{100}$$

$$\tag{101}$$

The following also holds:

$$\forall_{l_1 > l_2 \geq k}, \ C_{l_1,l_2} = \mathbb{E}\left((\sum_{l=k}^{l_1-1}y^{l\top}W^{l_1,l}Z^{l_1})^2\|y^{l_2}\|^2\right) = \frac{\alpha}{l_1}\sum_{l=k}^{l_1-1}C_{l,l_2} \tag{102}$$

and so:

$$C_{L,L} \leq \frac{(n+5+|\Delta|)\alpha^2}{nL^2}\Big(C_{L-1,L-1} + \frac{(L-1)^2 n}{\alpha^2(n+5-|\Delta|)}C_{L-1,L-1}... \tag{103}$$

$$+ \frac{2\alpha}{L-1}\sum_{l_1=k}^{L-2}\sum_{l_2=k}^{L-2}C_{l_1,l_2}\Big) \tag{104}$$

$$= \frac{(n+5+|\Delta|)\alpha^2}{nL^2}\Big(C_{L-1,L-1} + \frac{(L-1)^2 n}{\alpha^2(n+5-|\Delta|)}C_{L-1,L-1} \tag{105}$$

$$+ \frac{2n(L-1)}{\alpha(n+5-|\Delta|)}C_{L-1,L-1}\Big) \tag{106}$$

$$= \frac{(n+5+|\Delta|)\alpha^2}{nL^2}C_{L-1,L-1}\Big(1 + \frac{(L-1)^2 n}{\alpha^2(n+5-|\Delta|)} + \frac{2n(L-1)}{\alpha(n+5-|\Delta|)}\Big) \tag{107}$$

$$= C_{L-1,L-1}\frac{(n+5+|\Delta|)}{(n+5-|\Delta|)}\Big(\Big(\frac{\alpha}{L} + \frac{\alpha(L-1)}{L\alpha}\Big)^2 + \frac{\alpha^2(5-|\Delta|)}{nL^2}\Big) \tag{108}$$

$$C_{L,L} \leq C_{L-1,L-1}\Big(1 + \frac{2|\Delta|}{n+5-|\Delta|}\Big)\Big(\Big(1 + \frac{\alpha-1}{L}\Big)^2 + \frac{\alpha^2(5-|\Delta|)}{nL^2}\Big) \tag{109}$$

$$= C_{L-1,L-1}\Big(1 + \frac{2|\Delta|}{n+5-|\Delta|}\Big)\Big(1 + \frac{\alpha^2(5-|\Delta|)}{n(L+\alpha-1)^2}\Big)\Big(1 + \frac{\alpha-1}{L}\Big)^2 \tag{110}$$

Telescoping through $l = L-1...1$:

$$C_{L,L} \leq \Big(1 + \frac{2|\Delta|}{n+5-|\Delta|}\Big)^{L-k+1}\prod_{l\neq k}^{L}\Big(\Big(1 + \frac{\alpha^2(5-|\Delta|)}{n(l+\alpha-1)^2}\Big)\Big(1 + \frac{\alpha-1}{l}\Big)^2\Big) \tag{111}$$

$$\leq \exp\Big[\frac{2|\Delta|(L-1)}{n+5-|\Delta|}\Big]\exp\Big[\sum_{l\neq k}\frac{\alpha^2(5-|\Delta|)}{n(l+\alpha-1)^2}\Big]\exp\Big[\sum_{l\neq k}\frac{2(\alpha-1)}{l}\Big] \tag{112}$$

$$\sim \exp\Big[\frac{2|\Delta|(L-1)}{n+5-|\Delta|}\Big]\exp\Big[\frac{\alpha^2(5-|\Delta|)}{nL}\Big]L^{2(\alpha-1)} \tag{113}$$

$$\sim C_1\exp\Big[\frac{2|\Delta|L}{n}\Big]L^{2(\alpha-1)} \tag{114}$$

A similar relation holds for the mean:

$$\mu_L = \frac{\alpha}{L}\sum_{l=k}^{L-1}\mu_l = \frac{\alpha}{L}\Big(\mu_{L-1,L-1} + \frac{(L-1)}{\alpha}\Big)\mu_{L-1,L-1}\Big) \tag{115}$$

$$\mu_L^2 = \mu_{L-1,L-1}^2\Big(1 + \frac{\alpha-1}{L}\Big)^2 \leq L^{2(\alpha-1)} \tag{116}$$

Finally:

$$var(\|y^L\|^2) = C_{L,L} - \mu_L^2 \tag{117}$$

$$\leq L^{2(\alpha-1)}\Big(C_1\exp\Big[\frac{2|\Delta|L}{n}\Big] - 1\Big) \tag{118}$$

A lower bound can easily be derived using Eq. 98, 99, and follow the same derivation by flipping the sign of $|\Delta|$:

$$L^{(\alpha-1)}\Big(C_1\exp\Big[\frac{-2|\Delta|L}{n}\Big] - C_2\Big) \leq var(\|y^L\|^2) \leq L^{2(\alpha-1)}\Big(C_3\exp\Big[\frac{2|\Delta|L}{n}\Big] - C_4\Big) \tag{119}$$

For Gaussian distributions, using a similar derivation and assigning $\Delta = 0$ we have:

$$C_{L,L} = \prod_{l \neq k}^{L} \left( \left( 1 + \frac{5\alpha^2}{n(l+\alpha-1)^2} \right) \left( 1 + \frac{\alpha-1}{l} \right)^2 \right) \left( \frac{n+5}{nk^2} \right) a^2 \tag{120}$$

$$\mu_L^2 = \prod_{l \neq k}^{L} \left( 1 + \frac{\alpha-1}{l} \right)^2 \frac{a^2}{k^2} \tag{121}$$

and so:

$$var(\|y^L\|^2) = \left( a^2 \left( \frac{n+5}{nk^2} \right) \prod_{l \neq k}^{L} \left( 1 + \frac{5\alpha^2}{n(l+\alpha-1)^2} \right) - \frac{a^2}{k^2} \right) \prod_{l \neq k}^{L} \left( 1 + \frac{\alpha-1}{l} \right)^2 \tag{122}$$

### 7.6 PROOF OF LEMMA 3

The proof is identical to the Relu case, removing the activation variables $Z^l$, and hence will be omitted.

### 7.7 PROOF OF LEMMA 4

*Proof.* We construct the matrix $W^l$ as follows:

$$w_{ij}^l = \begin{cases} w_{ij}^{l1} & y^{l-1} \geq 0 \\ w_{ij}^{l2} & y^{l-1} < 0 \end{cases} \tag{123}$$

And so $y^l = W^{l\top} y^{l-1}$. Note that the elements of $W^l$ are i.i.d Gaussian random variables $w_{ij}^l \sim \mathcal{N}(0, \frac{\alpha}{n_{l-1}})$. □

## 8 EMPIRICAL SUPPORT FOR CONCATENATIVE RELU

Our theoretical results are verified in Fig. 3. As can be seen, the simulations closely match the predictions. The advantage of CR over conventional ReLU networks is also clearly visible.

CR has fallen out of flavor and was never shown to be effective outside CNNs. We, therefore, provide empirical support for its effectiveness, to which our analysis above points. In our experiments, we evaluate the performance of fully connected networks on the 40 UCI datasets with more than 1000 samples. The train/test splits were provided by Klambauer et al. (2017). In the experiment, we tested three different depths: 8, 16, and 32. Each layer contained $n = 256$ hidden neurons. Dropout was not employed. A learning rate of either 0.1, 0.01, or 0.001 was used for the first 50 epochs and then a learning rate of one tenth for an additional 50 epochs. All runs were terminated after 100 epochs. Batches were of size $N = 50$.

Following Klambauer et al. (2017), an averaging operator with a mask size of 5 was applied to the validation error, and the epoch with the best smoothed validation error was selected. This was done separately for each method, each dataset, and each split. Out of the nine options of depth and learning rate for most methods, the mean validation error across the four splits was used to select the option for which we report the results on the test splits. We compared the following activation functions and architectures: vanilla neural networks (single pathway) with either SELU (Klambauer et al., 2017) or ELU (Clevert et al., 2015) activation functions, ResNets, and CR based architecture. The single pathway ReLU networks were not competitive and the experiments with this architecture were terminated to allow for more experiments with the other architectures. The single pathway experiments were done without batchnorm, as is usually done with ELU and SELU. For ResNets, we report results both with batchnorm and without. Batchnorm was applied along the residual pathway, twice, as is typically practiced. The first and last layers of the ResNet and CR architecture, i.e., the layers that project from the input and to the output layer, are conventional projections that do not employ multiple pathways. For the baseline architectures, we also evaluate using double width, i.e.,

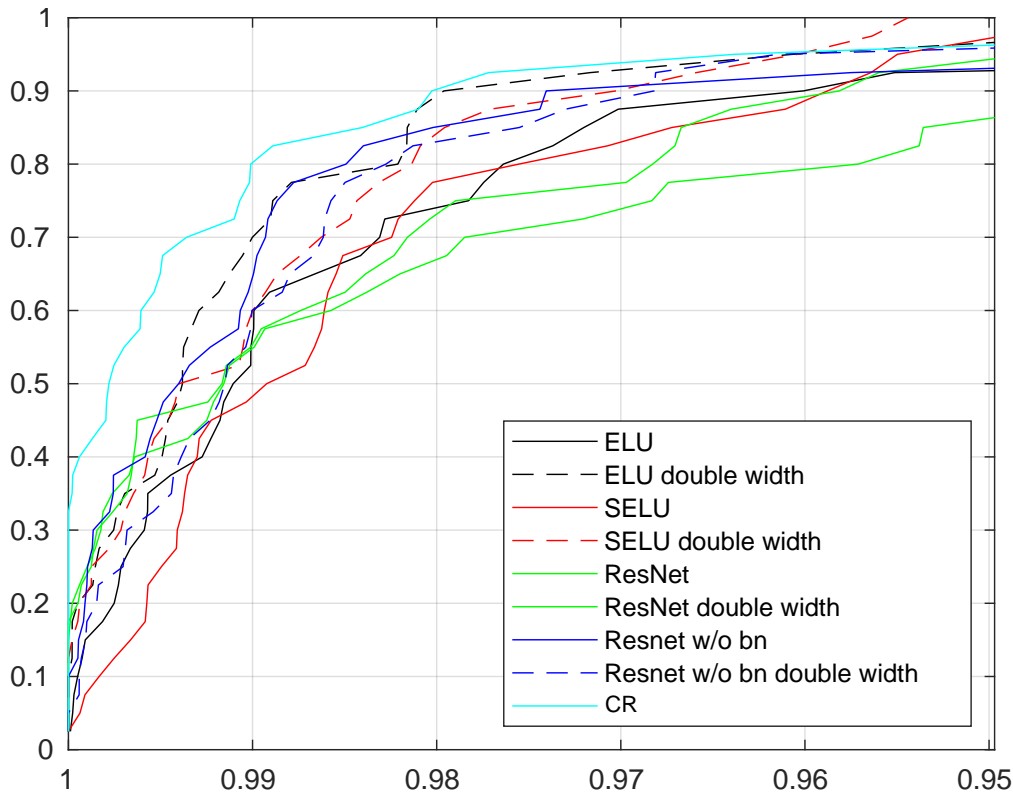

Figure 4: A Dolan-More profile, based on the accuracy obtained for the UCI experiments. The x-axis is the threshold ($\tau$). Note that since for accuracy scores, higher is better, whereas typical Dolan-More plots show cost, the axis is ordered in reverse. The y-axis depicts, for a given architecture out of the nine, the ratio of datasets in which the obtained accuracy is above the threshold $\tau$ times the maximal accuracy obtained by any architecture .

Table 1: Error rates for each method per dataset. 512 denote the double width settings.

| | CR | ELU | ELU 512 | SELU | SELU 512 | RES | RES 512 | RES w/o bn | RES 512 w/o bn |
|---|---|---|---|---|---|---|---|---|---|
| semeion | 0.102 | 0.1357 | 0.118 | 0.1543 | 0.1356 | 0.1192 | 0.1215 | 0.121 | 0.1308 |
| oocytes..._2f | 0.0941 | 0.0918 | 0.0949 | 0.0938 | 0.0892 | 0.0897 | 0.0921 | 0.0944 | 0.1084 |
| wine-quality-white | 0.4532 | 0.4633 | 0.4488 | 0.4597 | 0.4547 | 0.4521 | 0.4552 | 0.4573 | 0.4645 |
| ringnorm | 0.0208 | 0.0253 | 0.0251 | 0.026 | 0.0222 | 0.0261 | 0.0220 | 0.0205 | 0.0341 |
| statlog-image | 0.0448 | 0.0483 | 0.0472 | 0.042 | 0.0495 | 0.0467 | 0.0639 | 0.0489 | 0.0382 |
| cardiotocography-10 | 0.2090 | 0.2207 | 0.212 | 0.2222 | 0.2073 | 0.2147 | 0.2134 | 0.2126 | 0.2141 |
| abalone | 0.3408 | 0.3439 | 0.3391 | 0.3476 | 0.3492 | 0.3452 | 0.3479 | 0.3464 | 0.344 |
| ozone | 0.0295 | 0.0300 | 0.0287 | 0.0339 | 0.0288 | 0.0287 | 0.0312 | 0.0288 | 0.0321 |
| thyroid | 0.0207 | 0.0179 | 0.0183 | 0.0211 | 0.0198 | 0.0192 | 0.0186 | 0.018 | 0.0247 |
| oocyte..._4d | 0.1855 | 0.1946 | 0.194 | 0.2052 | 0.2076 | 0.2021 | 0.2006 | 0.2047 | 0.2156 |
| adult | 0.1515 | 0.1539 | 0.1530 | 0.1559 | 0.1550 | 0.1515 | 0.1509 | 0.1507 | 0.1538 |
| chess-krvkp | 0.0215 | 0.0333 | 0.0257 | 0.0307 | 0.0255 | 0.0205 | 0.0262 | 0.0275 | 0.0178 |
| waveform | 0.1403 | 0.1419 | 0.1441 | 0.1468 | 0.1455 | 0.1355 | 0.1399 | 0.1377 | 0.1656 |
| plant-texture | 0.2614 | 0.3412 | 0.2991 | 0.3022 | 0.2777 | 0.2814 | 0.2704 | 0.2738 | 0.3296 |
| statlog-landsat | 0.1061 | 0.1025 | 0.1011 | 0.1011 | 0.0987 | 0.1053 | 0.1111 | 0.1069 | 0.0960 |
| steel-plates | 0.275 | 0.2933 | 0.2815 | 0.2832 | 0.2776 | 0.2961 | 0.2879 | 0.2881 | 0.2926 |
| statlog-vehicle | 0.2188 | 0.2115 | 0.2159 | 0.2213 | 0.2233 | 0.2198 | 0.2285 | 0.2164 | 0.2075 |
| cardiotocography-3 | 0.0967 | 0.1057 | 0.1022 | 0.1023 | 0.1008 | 0.0988 | 0.1053 | 0.0994 | 0.1055 |
| page-blocks | 0.0324 | 0.0367 | 0.0345 | 0.0367 | 0.0355 | 0.0356 | 0.0343 | 0.0376 | 0.0367 |
| wall-following | 0.1131 | 0.1232 | 0.1272 | 0.1269 | 0.1235 | 0.1204 | 0.1236 | 0.1190 | 0.1192 |
| miniboone | 0.0793 | 0.0751 | 0.0750 | 0.0790 | 0.0790 | 0.0849 | 0.0889 | 0.0772 | 0.2481 |
| statlog-shuttle | 0.0016 | 0.002 | 0.0017 | 0.0015 | 0.0015 | 0.0025 | 0.0025 | 0.0022 | 0.0017 |
| mushroom | 0.0004 | 0.0004 | 0.0003 | 0.0010 | 0.0003 | 0.0010 | 0.0008 | 0.0007 | 0.0001 |
| waveform-noise | 0.1464 | 0.1501 | 0.1520 | 0.1626 | 0.1576 | 0.1437 | 0.1421 | 0.7090 | 0.1705 |
| twonorm | 0.0231 | 0.027 | 0.0252 | 0.0261 | 0.0258 | 0.0236 | 0.0243 | 0.0241 | 0.0341 |
| connect-4 | 0.1442 | 0.1508 | 0.1493 | 0.1483 | 0.1442 | 0.1423 | 0.1431 | 0.1424 | 0.1628 |
| letter | 0.0488 | 0.0700 | 0.0650 | 0.0606 | 0.0583 | 0.0880 | 0.0778 | 0.0818 | 0.0485 |
| titanic | 0.2197 | 0.2148 | 0.2151 | 0.2170 | 0.2118 | 0.2149 | 0.2181 | 0.212 | 0.2149 |
| statlog-heart | 0.1844 | 0.1939 | 0.1808 | 0.2093 | 0.2113 | 0.1967 | 0.2016 | 0.1794 | 0.2276 |
| plant-margin | 0.2798 | 0.3330 | 0.2835 | 0.3258 | 0.2918 | 0.2772 | 0.2764 | 0.2951 | 0.3749 |
| contrac | 0.4676 | 0.4757 | 0.4716 | 0.4816 | 0.4852 | 0.4606 | 0.4622 | 0.4693 | 0.4889 |
| yeast | 0.4239 | 0.4205 | 0.4252 | 0.4333 | 0.4253 | 0.4141 | 0.4199 | 0.4431 | 0.4649 |
| bank | 0.1121 | 0.1114 | 0.1131 | 0.1180 | 0.1170 | 0.1084 | 0.1161 | 0.1099 | 0.1143 |
| hill-valley | 0.3667 | 0.3699 | 0.3774 | 0.3700 | 0.3554 | 0.3899 | 0.3657 | 0.3765 | 0.4062 |
| optical | 0.0299 | 0.0456 | 0.0408 | 0.0416 | 0.0417 | 0.0477 | 0.0475 | 0.0426 | 0.0366 |
| plant-shape | 0.4127 | 0.5439 | 0.4862 | 0.4300 | 0.4220 | 0.4593 | 0.4581 | 0.4509 | 0.4228 |
| led-display | 0.2906 | 0.2929 | 0.2827 | 0.2911 | 0.3060 | 0.2852 | 0.2856 | 0.2781 | 0.3093 |
| magic | 0.1293 | 0.1310 | 0.1331 | 0.1345 | 0.1344 | 0.1375 | 0.1348 | 0.1329 | 0.1332 |
| pendigits | 0.0388 | 0.0446 | 0.0434 | 0.0426 | 0.0432 | 0.0466 | 0.0483 | 0.0467 | 0.0439 |
| chess-krvk | 0.2452 | 0.2592 | 0.2426 | 0.2399 | 0.2284 | 0.2956 | 0.2848 | 0.2741 | 0.3289 |

each layer had 512 hidden neurons. This was done since CR employs more weights than the baselines due to the dual pathway architecture.

In Fig. 4, we provide the Dolan-More profile of the methods. In this plot, the x-axis is a threshold $\tau$ and the y-axis is the ratio, out of all datasets, in which the method has reached an accuracy that is $\tau$ times the best accuracy for the dataset. It can be seen that the CR method dominates across all threshold values, except for SELU double width at low values of $\tau$.

## 9 UCI RESULTS

In Tab. 1 we provide the full numbers of the UCI experiments in the paper.

