# OpenReview forum: "The Effect of Residual Architecture on the Per-Layer Gradient of Deep Networks"
_ICLR.cc/2020/Conference — Reject_

### Official Review · AnonReviewer1 · 2019-10-17
**Official Blind Review #1**

**Rating:** 6

**Review:**

This paper studies the effects of residual and dense net type connections on the moments of per-layer gradients at random initialization. In particular, using duality, bounds on the variance of the square norm of Jacobian (with respect to the randomness of random initialization) are derived for vanilla networks. By noticing that connections that bypass a specific layer does not affect the expected square norm of the gradients of a particular layer, the paper continue to characterize the bounds for residual networks and densely connected networks that have special skip connections. In particular, with properly chosen initialization scales for each layer, the architectures with skip connections can be initialized so that the gradient norm does not explode with increasing depth.

In the second part of the paper, the topic shifted to a bit unrelated topic, comparing the effects of ReLU vs concatenated ReLU blocks on the norm of layer activations.

The results in this paper seem to be useful for guidance on choosing proper initialization schemes for different architectures at different depths. I hope the paper could spend more efforts discussing the potential impact of their results. Moreover, some preliminary empirical studies could also help:

- how well do the bounds still hold after the optimization starts to run?
- does the proposed initialization scheme actually allow training of very deep neural networks?

Minor comments:

- please add x and y labels to the plots
- please add grid (as in Fig 4) to Fig 2 and 3. Currently it is especially hard to decide the slope for curves in Fig 2b.

**Experience Assessment:**

I have published one or two papers in this area.

**Review Assessment: Checking Correctness Of Derivations And Theory:**

I assessed the sensibility of the derivations and theory.

**Review Assessment: Checking Correctness Of Experiments:**

I assessed the sensibility of the experiments.

**Review Assessment: Thoroughness In Paper Reading:**

I read the paper at least twice and used my best judgement in assessing the paper.

---

> ### Author Response · Authors · 2019-11-12
> **Thank you for your supportive feedback.**
>
> Regarding behaviour after training has begun: our work does not explore the dynamics of training itself and its effect on the different norms, and we do not have any reason to believe that the bounds hold during the course of training. We do, however, believe these bounds give an indication as to when training  can effectively start, and give some intuition to the different pathological failures of different architectures, in terms of exploding and vanishing gradients.
>
> We thank the reviewer for pointing out various flaws in the paper presentation. Following the reviewer’s comment, we have fixed various typos. Figures 2+3 have been replaced, and the new images present a grid and proper axis values.

---

### Official Review · AnonReviewer2 · 2019-10-20
**Official Blind Review #2**

**Rating:** 1

**Review:**

The paper studies the mean and variance of the gradient norm at each layer for vanilla feedforward, ResNet and DenseNet, respectively, at the initialization step, which is related with Hanin & Ronick 2018 studying the mean and variance of forward activations. They show that ResNet and DenseNet preserve the variance of the layer gradient norm through depths. In comparison, for the vanilla feedforward network, although the mean of the gradient norm is preserved if  is properly initialized, the variance of the layer gradient norm increases over depths, which may explode or decay the gradient at deeper layers.

The result presented in the paper is interesting, and the theory and empirical verification have a good match. I  recommend the acceptance after the following points are well addressed.

1. The mean and variance of layer activation norm and gradient norm have been studied in the mean field literatures for example [1] and the paper does not have a good comparison with them.
2. The experiment of  CONCATENATIVE RELU is not convincing given the small tasks intertwined performance.

Minor presentation flaws:
1. The sentence in Abstract "This depth invariant result is surprising in light of the literature results that state that the norm of the layer’s activations grows exponentially with the specific layer’s depth." is not clear itself because of no relevant reference.
2. There are many typos such as "weather", "for of" in the paper. Please carefully correct them.
3. The values on the y-axis of Figure 2 and Figure3 are not correctly shown.

[1] Greg Yang and Samuel Schoenholz 2017. Mean Field Residual Networks: On the Edge of Chaos


#####after rebuttal period, I found a fatal error in the paper#####

In (13), the network output $y_t^L$ is decomposed into two parts $y^{L,k}_t$ and $\hat{y}^{L,k}_t$, where $y^{L,k}_t$ is composed of paths that go through weight matrix $W^k$ and $\hat{y}^{L,k}_t$ is composed of paths that skip weight matrix $W^k$.

In (14), the paper derives $J^k:=\frac{\partial y_t^L}{\partial W^k} = \frac{\partial y^{L,k}_t}{\partial W^k}$. However, in fact the other part $\hat{y}^{L,k}_t$ also has gradient with respect to  $W^k$ even if the path $\gamma_t$ does not go through $W^k$ because the activation $z_{\gamma_t}$ is a function of  $W^k$ .

With this error, all the following claims in the paper are wrong. Thus I vote to reject this paper.

One can do experiment to verify  that "Thm. 1 also reveals a surprising property of the gradients in general Relu networks. That is, when the weights are sampled from the same distribution in each layer, the gradient’s magnitude for each layer are equal in expectation, and depend only on the output statistics." is wrong for ResNet.

**Experience Assessment:**

I have published one or two papers in this area.

**Review Assessment: Checking Correctness Of Derivations And Theory:**

I carefully checked the derivations and theory.

**Review Assessment: Checking Correctness Of Experiments:**

I assessed the sensibility of the experiments.

**Review Assessment: Thoroughness In Paper Reading:**

I read the paper thoroughly.

---

> ### Author Response · Authors · 2019-11-12
> **Thank you for your supportive feedback.**
>
> We thank the reviewer for the support. Mean field approximations have been used extensively in past work to study activation statistics, both in the forward and backward pass of different architectures. Some mean field--based papers have been mentioned in the related work section of the submitted manuscript. However, following the reviews, we have added more references and further elaborate on the differences between our approach and past work (see also our answer to R3).
> In general, the mean-field work differs from ours in the type of approximations, and the results that are obtained. Approximation wise, mean field results are obtained by applying the central limit theorem when taking the width to infinity. With this technique,  results studying the effect of  width for finite-sized networks cannot be obtained. It is also assumed that the weights of the forward and backward pass are independent (see Axiom 3.1 in [2]).
> In addition, we would like to point out that work such as [2], study mean behaviour of expected quantities. That is, mean squared norms of forward and backward signals. The variance of these quantities, however, is effectively nulled by the infinite width approximation. Our work, in contrast, characterizes the effect of finite width, and provides a more complete characterization of the behaviour of forward and backward statistics in the small width regime. We have added further clarification in the related work section regarding this difference.
>
> We could not completely parse the sentence “The experiment of  CONCATENATIVE RELU is not convincing given the small tasks intertwined performance”. However, we understand that the reviewer is not convinced by the CR experiments. While we believe that the experiments are quite comprehensive, this is not the main focus of our work. Following the review and the remark of R3 on the relevancy of these results to our theoretical contribution, in the revised version we have moved the empirical support of CR to the supplementary.
>
> We thank the reviewer for pointing out various flaws in the paper presentation. Following the reviewer’s comment, we have removed the sentence “this depth invariant result….” altogether from the abstract. In addition, we have fixed the various typos pointed to us by the reviewer and have sent the draft for a new round of professional proofreading. Figures 2+3 have been replaced with proper axis values.

---

> ### Author Response · Authors · 2019-12-21
> **reply to the update made after the rebuttal period**
>
> Since the edit of the review appeared late in the process, we are only able to address it after the decisions are released. However, we would be grateful for a response.
>
> In Eq (13), the output y^L is broken into a sum of two terms, one that contains the weight W^k and one that does not. Both terms contain the binary activation variable z. Since the activation is binary, its gradient with respect to W^k is zero, and therefore Eq (14), as well as the rest of the derivation, is correct.
>
> About the suggestion to verify Thm. 1  empirically. In the paper, we provided such verifications for more "interesting" results. The requested verification is shown below. (The figures can be found here: https://imgur.com/a/7zoV2ih)
>
> Figure 1. The per-layer squared norm of the Jacobian and the squared norm of y^k divided by the second moment of the weights, for a 20 layer resent of depth 10,15,20 (bottom plot to top plot) and alpha = 0.1.  As predicted in theorem 1, there is a perfect match and the norms are identical across all layers.
>
> Figure 2. The per-layer squared norm of the Jacobian and the squared norm of y^k divided by the second moment of the weights, for a 20 layer resent of depth 15, and alpha = 0.1,0.2,0.3, bottom plot to top plot.  As predicted in theorem 1, there is a perfect match and the norms are identical across all layers.

---

### Official Review · AnonReviewer3 · 2019-10-23
**Official Blind Review #3**

**Rating:** 6

**Review:**

This paper analyzes the statistics of activation norms and Jacobian norms for randomly-initialized ReLU networks in the presence (and absence) of various types of residual connections. Whereas the variance of the gradient norm grows with depth for vanilla networks, it can be depth-independent for residual networks when using the proper initialization.

The main theoretical results stem from a norm propagation duality that allows computations of certain statistics using simplified architectures. As far as I know, this is a novel way of analyzing multi-pathway architectures, and the method and the implied results will be of interest to deep-learning theorists.

There are two types of empirical results, one to verify the theoretical predictions, and the other to see if those predictions translate into improved generalization performance on some real-world tasks. Figures 2 and 3 show good agreement with theory. I am less convinced by Fig 4. Unless I missed something, I thought the implication from the theoretical analysis was that multi-pathway models should behave better than single-pathway models at large depth, and for single-pathway models concatenated ReLU may have slightly better large-depth behavior than standard ReLU. Doesn't this suggest that for large depth DenseNet ~= ResNet > CR > ReLU? Is the good performance of CR supposed to be understood as supporting evidence for the theoretical analysis? Or is the point more like "the theory suggests CR might be interesting, and indeed it performs well, but we leave the details of generalization performance to a future study"? It would be important to specify the desired interpretation here.

But I think regardless of the interpretation, the empirical results are not strong enough to stand on their own as support for CR, especially since this type of architecture has been introduced and studied previously (c.f. also the looks-linear setup from [1]). Therefore my evaluation is mostly based on the theoretical contributions, which I think are themselves are probably sufficient for publication.

One main area for improvement is in the related work. A significant line of work is omitted that studies higher-order statistics of Jacobian matrices in the large-depth, infinite-width regime [2-8], and while this is at infinite width, it work for any non-linearity, so is an important point for comparison. Other kinds of initialization schemes are also relevant for comparison, e.g. [9-10].

[1] Balduzzi, David, et al. "The shattered gradients problem: If resnets are the answer, then what is the question?." Proceedings of the 34th International Conference on Machine Learning-Volume 70. JMLR. org, 2017.
[2] Pennington, Jeffrey, Samuel Schoenholz, and Surya Ganguli. "Resurrecting the sigmoid in deep learning through dynamical isometry: theory and practice." Advances in neural information processing systems. 2017.
[3] Pennington, Jeffrey, Samuel Schoenholz, and Surya Ganguli. "The emergence of spectral universality in deep networks." International Conference on Artificial Intelligence and Statistics. 2018.
[4] Hayase, Tomohiro. "Almost Surely Asymptotic Freeness for Jacobian Spectrum of Deep Network." arXiv preprint arXiv:1908.03901 (2019).
[5] Tarnowski, Wojciech, et al. "Dynamical Isometry is Achieved in Residual Networks in a Universal Way for any Activation Function." The 22nd International Conference on Artificial Intelligence and Statistics. 2019.
[6] Burkholz, Rebekka, and Alina Dubatovka. "Initialization of ReLUs for Dynamical Isometry." arXiv preprint arXiv:1806.06362 (2018).
[7] Xiao, Lechao, et al. "Dynamical Isometry and a Mean Field Theory of CNNs: How to Train 10,000-Layer Vanilla Convolutional Neural Networks." International Conference on Machine Learning. 2018.
[8] Chen, Minmin, Jeffrey Pennington, and Samuel Schoenholz. "Dynamical Isometry and a Mean Field Theory of RNNs: Gating Enables Signal Propagation in Recurrent Neural Networks." International Conference on Machine Learning. 2018.
[9]Sutskever, Ilya, et al. "On the importance of initialization and momentum in deep learning." International conference on machine learning. 2013.
[10] Le, Quoc V., Navdeep Jaitly, and Geoffrey E. Hinton. "A simple way to initialize recurrent networks of rectified linear units." arXiv preprint arXiv:1504.00941 (2015).

**Experience Assessment:**

I have published in this field for several years.

**Review Assessment: Checking Correctness Of Derivations And Theory:**

I assessed the sensibility of the derivations and theory.

**Review Assessment: Checking Correctness Of Experiments:**

I assessed the sensibility of the experiments.

**Review Assessment: Thoroughness In Paper Reading:**

I read the paper at least twice and used my best judgement in assessing the paper.

---

> ### Author Response · Authors · 2019-11-12
> **Thank you for your supportive feedback.**
>
> We greatly appreciate pointing us to the missing references. As pointed out by the reviewer, the missing references analyze properties of the input - output Jacobian of various network architectures at initialization at the regime of infinite width and depth. Specifically, the mean squared singular value, as well as the entire distribution of squared singular values of this Jacobian are analyzed in pursuit of dynamical isometry, a state in which the squared singular values of the input - output jacobian are concentrated around 1. Dynamical isometry is indeed a much stronger condition then the Jacobian norm, as it requires access to the entire spectrum. However, in previous work, this is done using various forms of infinite width approximations.
> Instead, our work focuses on finite depth corrections to the statistics of various quantities concerning the squared Frobenius norm of the full per-layer Jacobian (derivative of the output with respect to the weights). Our approach can, therefore, be seen as a complementary approach to analyzing the dynamics at the start of training. We emphasize that our results cannot be obtained using large width approximations, since we seek to characterize the effect of both depth and width on the Jacobian.
> We have updated the manuscript with references to the relevant prior work, as well as added some discussion in order to better position our work in relation to it.
>
> Indeed supporting CR is only a minor goal of our work. However, since we demonstrated an advantage of CR layers over standard ReLU layers from the initialization perspective, we felt it would be beneficial to compare these empirically to other methods in settings in which multi pathway architectures are not yet dominant. Following the feedback from the reviewers, we have decided to move this empirical section to the appendix.

---

### Author Response · Authors · 2019-11-12
**Post-review revision**

Following the comprehensive reviews we have updated the manuscript.

1. Many mean-field references were added.
2. Figures 2,3 were improved
3. We have corrected multiple typos and edited according to the reviewers’ comments.
4. We have moved the CR experiments to the appendix.
5. Theorem 2 has been fixed after a few typos were found in its presentation.

---

### Decision · Program_Chairs · 2019-12-19

**Decision:**

Reject

**Comment:**

This paper studies the statistics of activation norms and Jacobian norms for randomly-initialized ReLU networks in the presence (and absence) of various types of residual connections. Whereas the variance of the gradient norm grows with depth for vanilla networks, it can be depth-independent for residual networks when using the proper initialization.

Reviewers were positive about the setup, but also pointed out important shortcomings on the current manuscript, especially related to the lack of significance of the measured gradient norm statistics with regards to generalisation, and with some techinical aspects of the derivations. For these reasons, the AC believes this paper will strongly benefit from an extra iteration.